# Pof8 is a La-related protein and a constitutive component of telomerase in fission yeast

Diego J. Páez-Moscoso[1], Lili Pan[1], Rutendo F. Sigauke[1], Morgan R. Schroeder[1], Wen Tang[1,5] & Peter Baumann[1,2,3,4]

Telomerase reverse transcriptase (TERT) and the non-coding telomerase RNA subunit (TR) constitute the core of telomerase. Here we now report that the putative F-box protein Pof8 is also a constitutive component of active telomerase in fission yeast. Pof8 functions in a hierarchical assembly pathway by promoting the binding of the Lsm2-8 complex to telomerase RNA, which in turn promotes binding of the catalytic subunit. Loss of Pof8 reduces TER1 stability, causes a severe assembly defect, and results in critically short telomeres. Structure profile searches identified similarities between Pof8 and telomerase subunits from ciliated protozoa, making Pof8 next to TERT the most widely conserved telomerase subunits identified to date.

---

[1] Stowers Institute for Medical Research, Kansas City, MO 64110, USA. [2] Howard Hughes Medical Institute, Kansas City, MO 64110, USA. [3] Department of Molecular and Integrative Physiology, University of Kansas Medical Center, Kansas City, MO 66160, USA. [4] Institute of Developmental Biology and Neurobiology, Johannes Gutenberg University, 55099 Mainz, Germany. [5] Present address: RNA Therapeutics Institute, University of Massachusetts Medical School, Worcester, MA 01605, USA. Correspondence and requests for materials should be addressed to P.B. (email: peter@baumannlab.org)

I n most eukaryotes, the DNA component of telomeres is composed of short tandem repeat sequences maintained by the reverse transcriptase telomerase. Telomerase is a ribonucleo-protein (RNP) complex in which the RNA subunit (TR, TER1 in *Schizosaccharomyces pombe*) functions as a scaffold for the assembly of telomerase reverse transcriptase (TERT, Trt1 in *S. pombe*) and other protein subunits[1]. The RNA subunit also contains the template region for telomere repeat synthesis. In all species examined, telomerase RNA subunits are transcribed as precursors that then undergo a series of processing events to produce the mature form that is assembled into the active telo-merase complex. In *S. pombe*, the mature form of TER1 is ~1213 nucleotides in length[2,3] and ends just upstream of a 5′ splice site[4]. The precursor is about 200 nucleotides longer, containing an intron and second exon. Interestingly, TER1 maturation involves only the first step of a splicing reaction. After spliceosomal cleavage at the 5′ splice site, the first exon is released to become the mature form of telomerase. This reaction is favored by RNA elements within TER1 that promote a slow transition between the two steps of splicing resulting in the "discard" of splicing inter-mediates[5]. A role for spliceosomal cleavage in 3′ end processing of telomerase is conserved among many fungi, but the underlying mechanisms are surprisingly diverse[6,7].

Fungal telomerase RNAs contain an Sm-binding site near the mature 3′ end, and require binding of the hetero-heptameric Sm complex for processing and stability[4,8–10]. The Sm complex is a member of the Hfq-family of RNA-binding proteins that is conserved in all domains of life[11]. Sm proteins also assemble on spliceosomal small nuclear RNAs (snRNAs), where they are cri-tical for 5′ cap hypermethylation, reimport of the RNPs into the nucleus and spliceosome function[12]. During *S. pombe* TER1 maturation, Sm proteins promote spliceosomal cleavage and recruit the methyl transferase Tgs1 that generates the 2,2,7-tri-methylguanosine (TMG) cap[8]. The Sm complex then dissociates from TER1 and is replaced by the Sm-like complex Lsm2-8, which protects the 3′ end of TER1 from degradation and pro-motes the association of TER1 with Trt1 to generate the func-tional enzyme[8].

Biochemical and structural studies of telomerases from ciliated protozoa has provided fundamental insights into telomerase biogenesis and function. However, the extent to which these findings can inform studies in other organisms has remained less clear due to fundamental differences in enzyme composition and biogenesis pathways. For example, the telomerase RNA subunit is transcribed by RNA polymerase III in ciliates, but by RNA pol II in yeasts and metazoans. It has thus been thought that proteins involved in the processing and stabilization of pol III transcripts may only function in telomerase RNA biogenesis in ciliates. This includes members of the Lupus La antigen-related protein (LARP) family[13], which are components of the telomerase holoenzyme in ciliates and are critical for the assembly, nuclear retention, and activity of telomerase[14–18].

We now demonstrate that a critical role for La family members in telomerase biogenesis and function is conserved in fission yeast, where telomerase RNA is a pol II transcript. Our results reveal that the Pof8 protein binds to telomerase RNA, functions in hierarchical assembly by promoting Lsm2-8 binding, and forms a constitutive component of the active enzyme. Profile database searches identify Pof8 as a previously unrecognized member of the LARP family displaying striking structural similarities with human and ciliate proteins. Our findings reveal an ancient role for La-related proteins (LARP) in telomerase biogenesis and indicate that evolutionary conservation in holoenzyme composition extends much further than previously thought.

## Results

**Pof8 is a La-related protein family member.** To gain a better understanding of the transition from Sm protein-bound TER1 precursor to the Lsm-bound mature form, we performed immunoprecipitations for each of the Sm and Lsm proteins from strains with myc epitope tags on individual subunits[8]. Precipitates were used to identify associated proteins and RNAs by mass spectrometry and Illumina sequencing, respectively. Our atten-tion focused on the Pof8 protein as it was reliably precipitated by Lsm2 and Lsm8 and not found in control IPs. Originally reported as a putative F-box protein[19], Pof8 had previously been impli-cated in telomere maintenance by screening the *S. pombe* gene deletion collection for strains with abnormal telomere length[20]. Using sequence- and profile-based searches, we were unable to independently confirm the F-box domain previously described[21]. However, a homology search readily identified an RNA recog-nition motif (RRM) near the C-terminus of Pof8 (Fig. 1a, b). This RRM most closely resembled RRMs in the human LARP family. A subsequent profile sequence search of the full-length Pof8 sequence using HHpred[22] revealed a La motif and an additional RRM (Fig. 1a, c). Both domains independently iden-tified Pof8 as a LARP, the same family that includes the telo-merase subunits p65 from *Tetrahymena thermophila*[15] and p43 from *Euplotes aediculatus*[14].

**Reduced TER1 level and short telomeres in *pof8Δ* cells.** The sequence similarity with bona fide telomerase components in ciliates, combined with the interaction of Pof8 with Lsm proteins, lead us to hypothesize that Pof8 may be directly involved in telomerase biogenesis. Examination of TER1 by northern blotting revealed a four to five-fold reduction in RNA levels in *pof8Δ* cells (Fig. 2a). The reduction in steady-state level predominantly affected the mature form generated by spliceosomal cleavage, whereas the levels of precursor and spliced form were only slightly reduced (Fig. 2a, b). The levels of Smb1, Sme1, Lsm4, Lsm5 also remained unchanged in the absence of Pof8, and the level of Trt1 protein was only slightly reduced indicating that the reduction in the mature form of TER1 is a direct consequence of loss of Pof8 protein (Supplementary Fig. 1). Consistent with the observations of the genome-wide telomere length screen[20], we found telomere length to be very short in the absence of Pof8 (Fig. 2c). It is important to note that the chromosome terminal fragments following digest of genomic DNA with *Eco*RI are composed of ~800 bp subtelomeric DNA and a variable number of telomeric repeats. The difference in mobility between wildtype and *pof8Δ* is therefore indicative of critically short telomeres with most of the remaining fragment composed of subtelomeric DNA. Although telomeres were maintained at this short length over successive restreaks at the population level, a fraction of telomeres nevertheless became uncapped and chromosome end fusions were readily detected in *pof8Δ* cells, but not in wild-type cells under the same condition (Fig. 2d). A *ter1Δ* strain in crisis served as a positive control for chromosome end fusions.

**Pof8 deletion impairs telomerase activity**. The pronounced telomere defect observed here was difficult to reconcile with the modest reduction in telomerase RNA level observed by northern blotting. To investigate whether telomerase activity was affected more severely than expected from the four-fold reduction in TER1 RNA, we performed direct telomerase activity assays from Trt1 immunoprecipitates prepared from cell extracts of *pof8+* and *pof8Δ* cells. We observed a 20- to 30-fold reduction in telomerase activity in the absence of Pof8 (Fig. 2e). Even less telomerase activity was detected when telomerase was immunoprecipitated with Lsm proteins from *pof8Δ* cells (Fig. 2f).

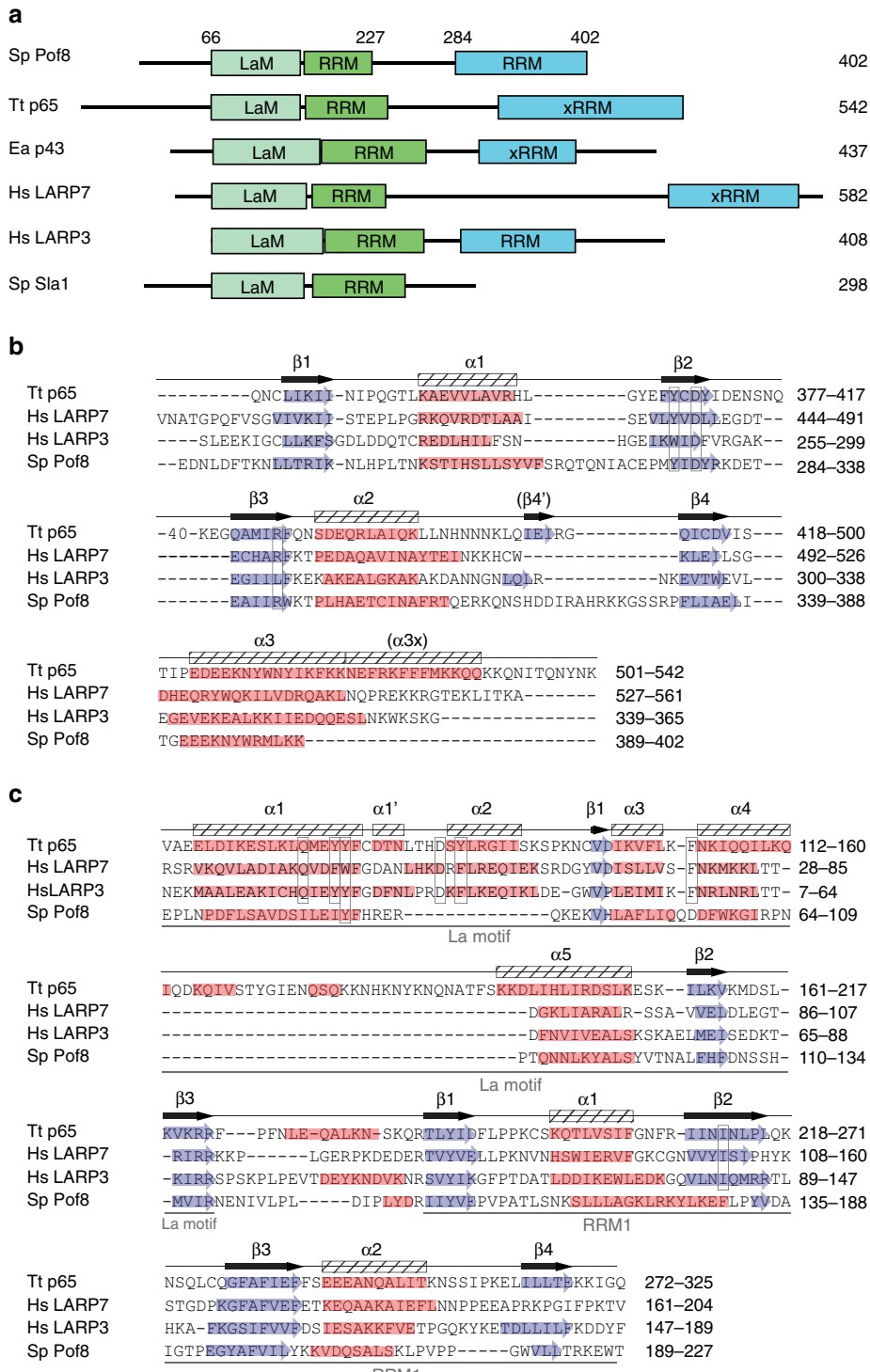

**Fig. 1** Pof8 structural analysis reveals a La motif and two RRM domains. **a** The domain architecture of *S. pombe* (Sp) Pof8 compared to *T. thermophila* (Tt) telomerase subunit p65, *E. aediculatus* (Ea) telomerase subunit p43, *Homo sapiens* (Hs) LARP7 and LARP3, and the *S. pombe* La protein, Sla1. Numbers above the Pof8 sequence indicate predicted domain boundaries. **b** Alignment and secondary structure elements of the C-terminal RRM domains based on HHPred predictions; beta strands are highlighted with blue arrows, and alpha helices are highlighted in red. The secondary structure elements for p65 are based on the crystal structure[27] and are labeled above the alignment. Forty amino acids that form an extended loop between β2 and β3 in p65 have been abbreviated -40- to keep the alignment compact. **c** N-terminal La motif and RRM1 domains. Residues shown to interact with oligo-uridine RNA are boxed and are mostly not conserved in Pof8. The p65 secondary structure elements are labeled above the alignment

To separate the effects of *pof8* deletion on TER1 stability from a role for Pof8 in telomerase biogenesis or regulation of activity, we expressed TER1 from the inducible *nmt1* promoter (Fig. 3a). In the induced state, TER1 levels were 10-fold higher in the *pof8* deletion strains (lanes 5 and 6) compared to wildtype (lane 1). Regardless of the amount of TER1, telomerase activity was dramatically reduced in the absence of Pof8 (Fig. 3b). Only after contrast adjusting beyond the point of saturation for the signal

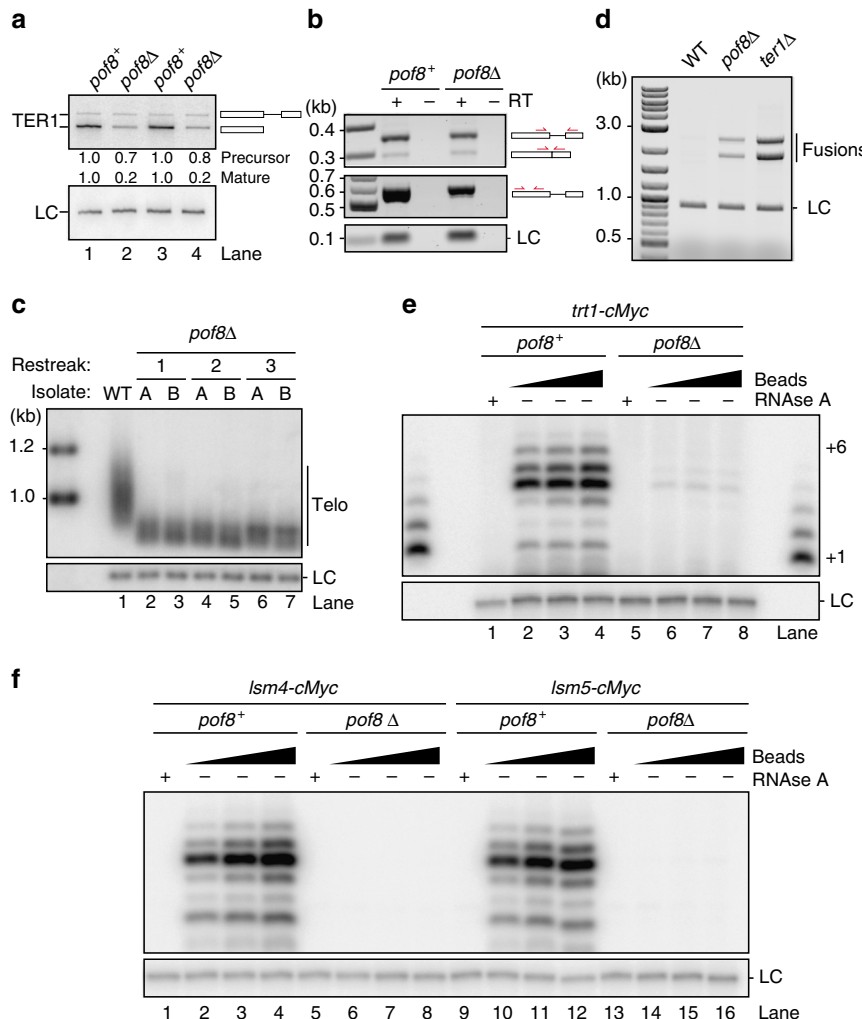

**Fig. 2** Telomerase and telomere defects associated with the deletion of *pof8*. **a** Northern blot analysis of TER1 from wildtype and *pof8Δ* strains following RNaseH cleavage. Quantification of TER1 signal relative to wildtype and normalized to the loading control (LC, snR101) is shown below the figure. Lanes 1 and 3 represent independent isolates for *pof8⁺*, and lanes 2 and 4 for *pof8Δ*. The schematics on the right represent TER1 precursor and mature forms. Uncropped blots are presented in Supplementary Fig. 3a. **b** RT-PCR amplification of spliced and unspliced TER1 (top panel), total TER1 (middle panel), and snRNA U1 used as loading control (LC, lower panel). The schematics on the right indicate the position of primers used relative to the structure of precursor, spliced and cleaved TER1 (RT reverse transcriptase). **c** Southern blot analysis to compare telomere length from wildtype (WT) and *pof8Δ* cells. Two independent isolates of *pof8Δ* (A and B) were restreaked three times and telomere length was analyzed (one restreak = 20–25 generations). The *rad16⁺* locus was probed as a loading control (LC). Lane numbers are indicated below the blot. Uncropped blot is presented in Supplementary Fig. 3a. **d** Detection of telomere–telomere fusions by PCR. **e** Telomerase activity assay from Trt1-cMyc immunoprecipitations. To establish RNA dependence, RNAse A was added to control samples prior to incubation with DNA primer and nucleotides. A ³²P-labeled 100-mer oligonucleotide was used as LC and a ladder of extension products generated by terminal transferase was run on both edges of the gel as size markers; the bands corresponding to 1 and 6 nucleotide addition products are labeled on the right. Uncropped blot is presented in Supplementary Fig. 3a. **f** As **e** but from strains harboring Lsm4-cMyc and Lsm5-cMyc in wildtype and *pof8Δ* backgrounds

from *pof8⁺* samples, weak telomerase activity was detected in the *pof8Δ* extracts (Fig. 3b, lower panel). Despite higher levels of TER1 RNA in the induced *pof8Δ* cells compared to the uninduced *pof8⁺* cells, the activity was over 300-fold reduced (compare lanes 2–4 with 14–16). Furthermore, despite a 16-fold higher level of telomerase RNA in induced *pof8Δ* cells vs. uninduced cells, the activity increased by less than two-fold. Consistent with the low activity despite overexpression of TER1, the short telomere phenotype of *pof8Δ* cells was not rescued (Fig. 3c). TER1 levels also increased following deletion of the RNA exonuclease *rrp6* (Supplementary Fig. 2a)[23]. In line with the results obtained by overexpressing TER1, deletion of *rrp6* also failed to rescue telomerase activity (Supplementary Fig. 2b). In summary, deletion of *pof8* caused a four-fold reduction in the steady-state

level of TER1, but had a far more dramatic effect on telomerase activity. Overexpressing TER1 or interfering with TER1 degradation rescued the RNA level, but failed to rescue telomerase activity and telomere shortening.

**Effect of *pof8* deletion on telomerase assembly**. The strong effect of *pof8* deletion on telomerase activity indicated that Pof8 may function as an assembly factor for telomerase. As shown previously[8], Lsm2-8 proteins are associated with the majority of mature telomerase RNA, whereas Sm proteins are bound to the precursor and a minor fraction of mature TER1. Deletion of *pof8* results in a 20-fold reduction in the amount of TER1 immuno-precipitated with Lsm4 and Lsm5, and a slight increase in Smb1

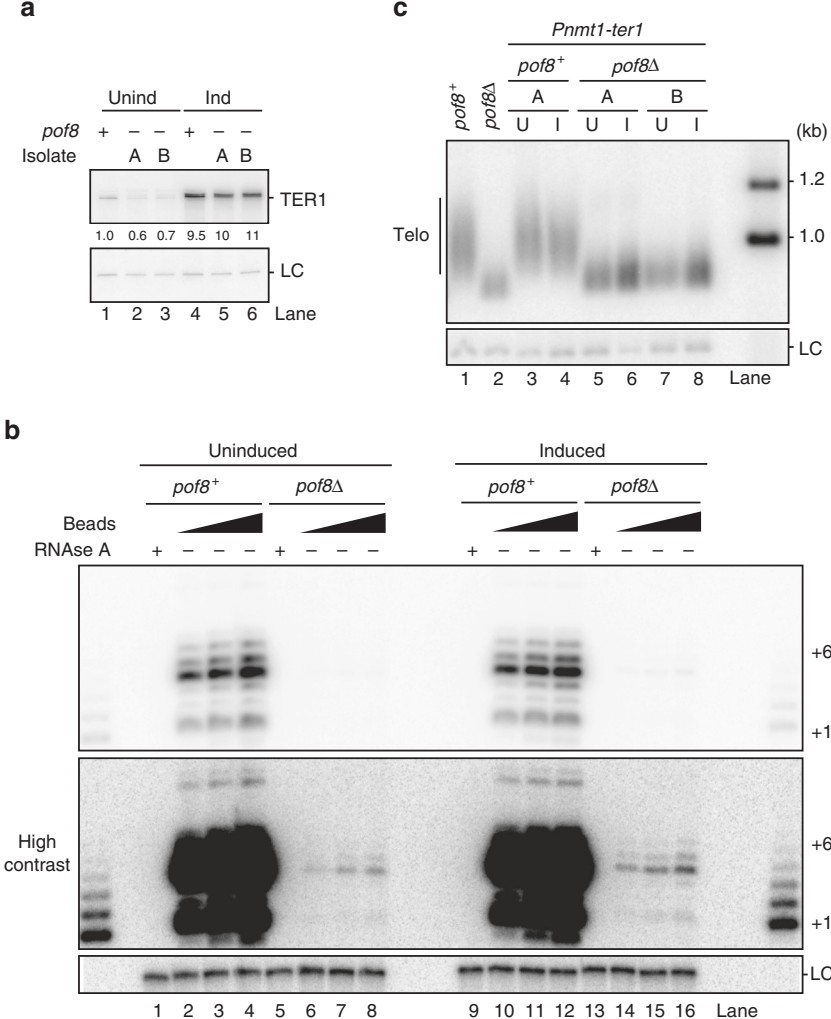

**Fig. 3** TER1 overexpression fails to rescue telomerase defects in *pof8Δ* cells. **a** Northern blot analysis of total RNA isolated from cells expressing TER1 from a plasmid under the control of uninduced (Unind) and induced (ind) nmt1 promoter in an Lsm4-cMyc background. Two independent isolates (A and B) of *pof8Δ* transformants were used. Uninduced cells were grown in YES (with thiamine) and in EMM (no thiamine) for the induced condition. The lower panel shows snR101 probed as a loading control (LC). Uncropped blots are presented in Supplementary Fig. 3b. **b** Telomerase activity assay from Lsm4-cMyc immunoprecipitates from *pof8⁺* and *pof8Δ* cells harboring nmt1-TER1 grown under uninduced and induced conditions. A high contrast version of the top panel is included to visualize the low levels of telomerase activity from *pof8Δ* cells. Ladder of extension products generated by terminal transferase is flanking the assay as size markers; the bands corresponding to +1 and +6 nucleotide addition products are indicated. A $^{32}$P-labeled 100-mer oligonucleotide was used as LC. Uncropped blot is presented in Supplementary Fig. 3b. **c** Southern blot analysis of telomere length of strains described in **a** and controls. U uninduced, I induced. A probe for the *rad16⁺* locus was used as the LC

and Sme1 association (Fig. 4a). These results indicate that loading of Lsm proteins onto TER1 is compromised in the absence of Pof8. To exclude the possibility that the reduction in TER1 level caused by the deletion of *pof8* was responsible for the diminished recovery by immunoprecipitation, we repeated the experiment in the context of overexpressed TER1 (Fig. 4b). Whereas immunoprecipitation of Lsm4 from a *pof8⁺* strain depleted over 50% of TER1 from the supernatant (lane 3), no measurable depletion was observed in the *pof8Δ* extract (lane 4, compare input with S/N) and TER1 was barely detectable in the immunoprecipitate (IP, lane 4, lower panel). We conclude that Lsm association with TER1 is compromised in the absence of Pof8. As we have previously shown that Trt1 association with TER1 requires prior binding of Lsm2-8[8], these experiments place Pof8 upstream of Lsm and Trt1 in the hierarchical assembly of telomerase and explain the dramatic reduction in telomerase activity in the absence of Pof8.

We next wanted to know whether Pof8 directly and stably associates with TER1. Following introduction of an N-terminal 3xFLAG epitope tag, Pof8 was detected as a single band by western blotting (Fig. 5a). Telomeres continued to be maintained at near wildtype length indicating that the 3xFLAG tag has little effect on Pof8 function (Fig. 5b). To test whether Pof8 binds directly to TER1, we incubated a radiolabeled probe corresponding to the short arms of TER1 with *S. pombe* extract containing FLAG-tagged Pof8 and UV irradiated to crosslink RNA–protein interactions. Pof8 was then immunoprecipitated under denaturing conditions to disrupt indirect interactions. Parallel control experiments were carried out with extracts containing no tags and myc epitope-tagged Lsm4, respectively. RNA was isolated from each IP, separated by gel electrophoresis and visualized (Fig. 5c). TER1 was found to be 2.5-fold enriched in the Lsm4 and Pof8 IPs relative to the untagged control, strongly supporting that Pof8 directly interacts with TER1, as does Lsm4.

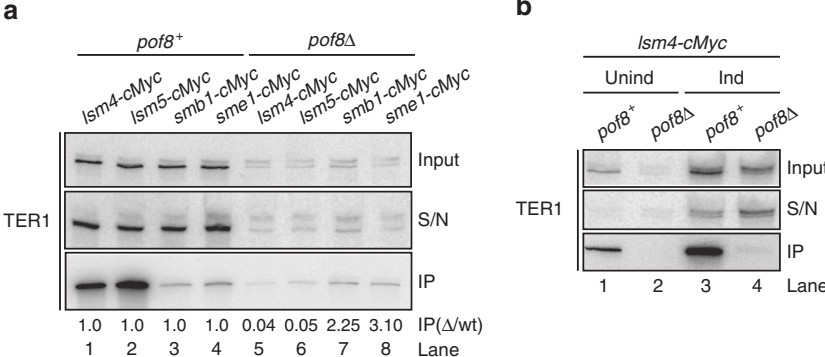

**Fig. 4** Lsm2-8 complex fails to associate with TER1 in Pof8-deficient cells. **a** Northern blot analysis of TER1 following immunoprecipitation (IP) of cMyc-tagged Lsm and Sm proteins. Input and supernatant (S/N) represent 10% of the IP samples. Upper band corresponds to precursor, lower band to mature form. IP signal was normalized to precursor plus mature form from the input samples and fold enrichment is shown relative to wildtype below the blots. The uncropped blots are presented in Supplementary Fig. 3c. **b** Northern blot analysis of TER1 from anti-cMyc IP samples. Strains contain plasmid borne, inducible TER1 in an Lsm4-cMyc background. Input and S/N represent 10% of the IP samples (Unind uninduced, Ind induced). The uncropped blots are presented in Supplementary Fig. 3c

**Pof8 is a subunit of active telomerase**. To assess whether Pof8 stably associates with TER1 in the context of active enzyme, we analyzed FLAG-Pof8 immunoprecipitations carried out under native conditions by northern blotting. TER1 was readily detected in immunoprecipitates from tagged, but not from control extracts in which Pof8 was untagged (Fig. 6a). The snRNA snR101 was not precipitated by Pof8 and thus served as specificity control. Whereas no activity was observed in control samples, FLAG-Pof8 immunoprecipitates displayed robust telomerase activity (Fig. 6b). To assess more quantitatively what fraction of telomerase is Pof8-associated, we generated strains containing Lsm4-cMyc in combination with FLAG-tagged or untagged Pof8. Cell-free extracts from these strains were subjected to a first round of immunoprecipitation using anti-FLAG antibody. The supernatant was then incubated with anti-cMyc to precipitate Lsm4-associated TER1. The four immunoprecipitates were then assayed for telomerase activity. As expected, no telomerase activity was precipitated in the FLAG IP from extract containing untagged Pof8 (Fig. 6c, lanes 2–4). Subsequent IP of Lsm4-cMyc from the supernatant of the first-round IP recovered robust activity (lanes 6–8). In contrast, when Pof8-associated telomerase was first precipitated (lanes 10–12), only 1% of activity was recovered in a subsequent Lsm4 IP (lanes 14–16). Based on these results, nearly 100% of active telomerase are associated with Pof8, making this protein a bona fide component of the active holoenzyme.

**Pof8 is not a general loading factor for Lsm2-8**. Considering the homology with La-related proteins and the stable association with telomerase, we wondered whether other RNAs were also affected by deletion of *pof8*. Using ribo-depleted RNA from otherwise isogenic *pof8+* and *pof8Δ* strains, we performed expression analysis in triplicate. Five protein-encoding transcripts and 13 non-coding RNAs including TER1 were found to be expressed at more than two-fold lower levels in *pof8Δ* cells (Supplementary Table 3a). An even smaller number of transcripts was found to be upregulated by greater than two-fold (Supplementary Table 3b). Among the seven upregulated protein-encoding genes was tlh2, a locus located in subtelomeric DNA and previously found to be upregulated in cells with critically short telomeres[24].

Although U6 snRNA associates with the Lsm2-8 complex like TER1, the U6 expression level was unaffected by the presence or absence of Pof8. Furthermore, immunoprecipitates from FLAG-Pof8 extracts contained barely more U6 than control IPs from extracts with untagged Pof8 (Fig. 7a). This argues against Pof8

being a general loading factor for Lsm2-8 and a component of the U6 snRNP. To further assess the specificity of Pof8 for loading the Lsm complex onto RNAs, we asked how many other RNAs that are associated with Lsm8 are affected in abundance by deletion of *pof8*. We chose Lsm8 for this experiment as it is the only Lsm family member unique to the Lsm2-8 complex, all others being shared by the cytoplasmic Lsm1-7 complex. Immunoprecipitation of Lsm8-cMyc enriched 35 RNAs by greater than two-fold, including U6 (30-fold enrichment) and TER1 (159-fold enrichment) (Supplementary Table 4). Overlaying the differential expression data for *pof8* with the Lsm8 IP revealed TER1 as the only transcript that is bound by Lsm8 and reduced in *pof8Δ* cells (Fig. 7b).

## Discussion

In summary, our results demonstrate that Pof8 is a previously unrecognized member of the LARP family that shares structural and functional similarity with telomerase subunits from ciliated protozoa, namely p43 and p65. Pof8 binds to telomerase RNA and promotes the loading of the Lsm2-8 complex, which in turn promotes the loading of the catalytic subunit Trt1. Pof8 is associated with nearly all telomerase activity, establishing the protein as a bona fide protein component of functional telomerase in fission yeast.

The identification of La-related proteins in association with highly purified telomerase from *Euplotes* and *Tetrahymena* was initially seen as unsurprising due to the nature of these RNAs as pol III transcripts. La protein associates rapidly and in most cases transiently with nascent pol III transcripts, and guides the RNAs through various processing steps[25]. Interestingly though, subsequent studies revealed that p43 and p65 are telomerase-specific proteins, suggesting functions distinct from those provided by the canonical La proteins[15,17]. A series of elegant biochemical[15,26], structural[27,28], and single-molecule[29] experiments have since demonstrated that p65 binding to telomerase RNA induces structural changes in the RNA that are instrumental for the hierarchical assembly of functional telomerase in *Tetrahymena*.

Much less is known about the assembly pathways for telomerases from other organisms, but recent findings have revealed more differences than commonalities. Budding yeast TLC1 is stably associated with Sm proteins[9], whereas fission yeast TER1 undergoes the Sm to Lsm switch[8]. While divergent yeast telomerase RNAs share features with snRNAs, vertebrate TRs belong to the family of H/ACA box snoRNAs[30] and scaRNAs[31], and

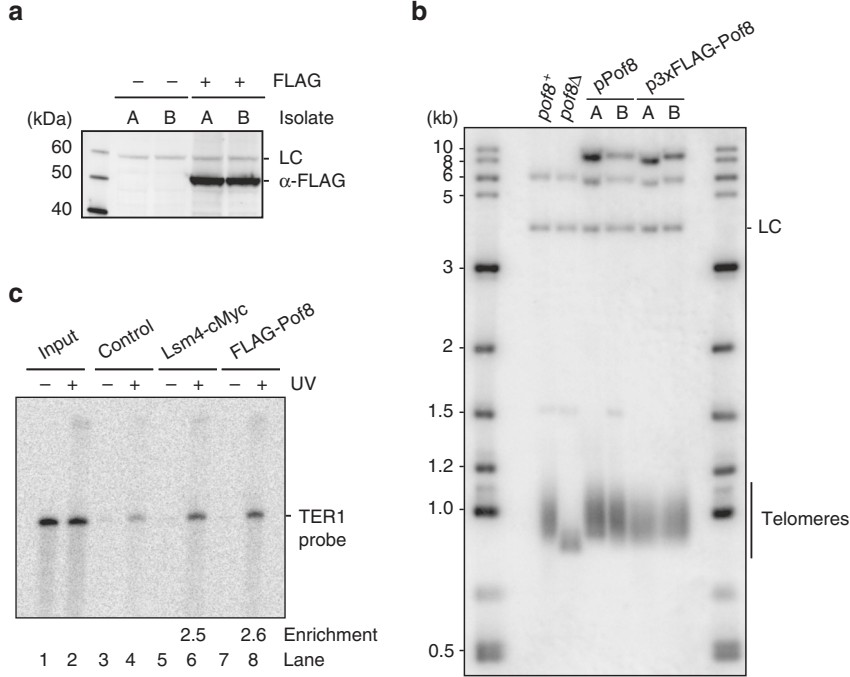

**Fig. 5** Characterization of 3xFLAG-Pof8. **a** Western blot using α-FLAG antibody in strains containing untagged Pof8 or 3xFLAG-Pof8 with two independent isolates per strain in an Lsm4-cMyc background. A non-specific band recognized by α-FLAG in *S. pombe* extracts even in the absence of an epitope-tagged protein was used as a loading control (LC). **b** Telomeric Southern blot from strains containing untagged Pof8 or 3xFLAG-Pof8 on a plasmid under the control of its endogenous promoter with two independent isolates per strain in an Lsm4-cMyc background. A *rad16*+ probe was used as the LC. **c** UV crosslinking and immunoprecipitation of in vitro transcribed TER1 probe incubated in extracts containing epitope-tagged Lsm4 or Pof8

associate with a different set of proteins to mediate 3′ end processing and stability[32–35]. The characterization of several telomerase components that lack clear functional orthologs in other eukaryotes supports the view that telomerases evolve far more rapidly than other well-characterized RNPs, and that the presence of a reverse transcriptase and a highly divergent non-coding RNA may be the only common denominators. Along these lines, it was recently shown that Pop1, 6, and 7, previously characterized as binding the P3 domains of RNaseP and MRP, are also constitutive components of telomerase in budding yeast and are critical for holoenzyme integrity in this organism[36,37]. Whether this remarkable feature of a conserved functional module being shared between three highly divergent ancient RNPs is conserved in other species remains to be seen. It appears that telomerase RNA easily acquires functional sequence modules, such as the P3 or H/ACA domains, and that active telomerase can be produced via diverse pathways.

The identification of a La-related protein as a constitutive telomerase subunit in fission yeast presents a different perspective. This is the first telomerase subunit since the cloning of the catalytic subunits in the late 1990's that is structurally and functionally conserved between organisms as distant as ciliates and yeasts. Despite a nearly 10-fold difference in size between ciliate and yeast TRs, a La-related protein has now been found to function in telomerase biogenesis in a seemingly conserved manner. Notably, two human La family members, LARP3 and LARP7, have also been implicated in telomere maintenance. LARP7 has been reported to affect alternative splicing of Tert[38], whereas LARP3 was found to associate with telomerase RNA[39]. Whether either of these proteins functions in a similar manner to p65 and Pof8 to promote telomerase assembly and activity remains to be tested. Studies of p65 were instrumental in establishing the paradigm of hierarchical assembly of functional telomerase aided by RNA chaperones[40]. In *S. pombe*, Sm proteins

assemble on the precursor of TER1, promote spliceosomal cleavage, and recruit Tgs1, which hypermethylates the mono-methyl guanosine cap. Through the present analysis, a picture of an ordered sequence of events emerges: Pof8 must bind to TER1 either before or immediately after spliceosomal cleavage, possibly triggering the departure of Sm proteins. Pof8 binding is then a prerequisite for Lsm2-8 loading onto the 3′ end of TER1. This may involve the recruitment of additional factors and/or a Pof8-induced conformational change in the RNA that creates a high-affinity Lsm-binding site. Pof8 remains associated with the active holoenzyme even after Lsm2-8, and subsequently Trt1, bind to TER1. The requirement for Pof8 for the Lsm–TER1 association and the formation of a Pof8–Lsm–Trt1–TER1 complex now suggests that hierarchical assembly is a conserved path for telomerase biogenesis that may also have a place in telomerase assembly in metazoans.

## Methods

**Strains and constructs**. *S. pombe* strains used in this study are listed in Supplementary Table 1. The *pof8* deletion was generated by replacing the complete open reading frame with the kanamycin resistance cassette using standard laboratory techniques[41,42]. Knockout fragments contained ~750 base pair (bp) upstream and downstream homology, and were generated by fusion PCR using primers listed in Supplementary Table 2. Cells were grown to late log phase and transformed by lithium acetate method. Transformants were selected on YEA plates plus geneticin disulfate (100 μg ml⁻¹). Epitope tags were introduced following the same strategy. Other strains were generated by crossing and selection of correct genotypes. The 3xFLAG tag was introduced at the N-terminus of Pof8 by fusion PCR in the context of a genomic fragment encompassing sequence from position −386 to +1769 and cloned into pDBlet plasmid[43]. All strains were verified by PCR or western blotting. Plasmids were introduced into *S. pombe* cells by electroporation.

**Telomere length analysis and fusion assay**. DNA preparation and telomere length analysis were performed based on ref. [44]. Cells from 20 ml cultures (~1 × 10⁹ cells per ml) were incubated with 2 ml of Z buffer (50 mM sodium citrate, 50 mM sodium phosphate dibasic, and 40 mM EDTA pH 7.8) plus 0.5 mg ml⁻¹ Zymolase T100 (US Biological) and 2 mM dithiothreitol (DTT) for 1 h at 37 °C. Sodium

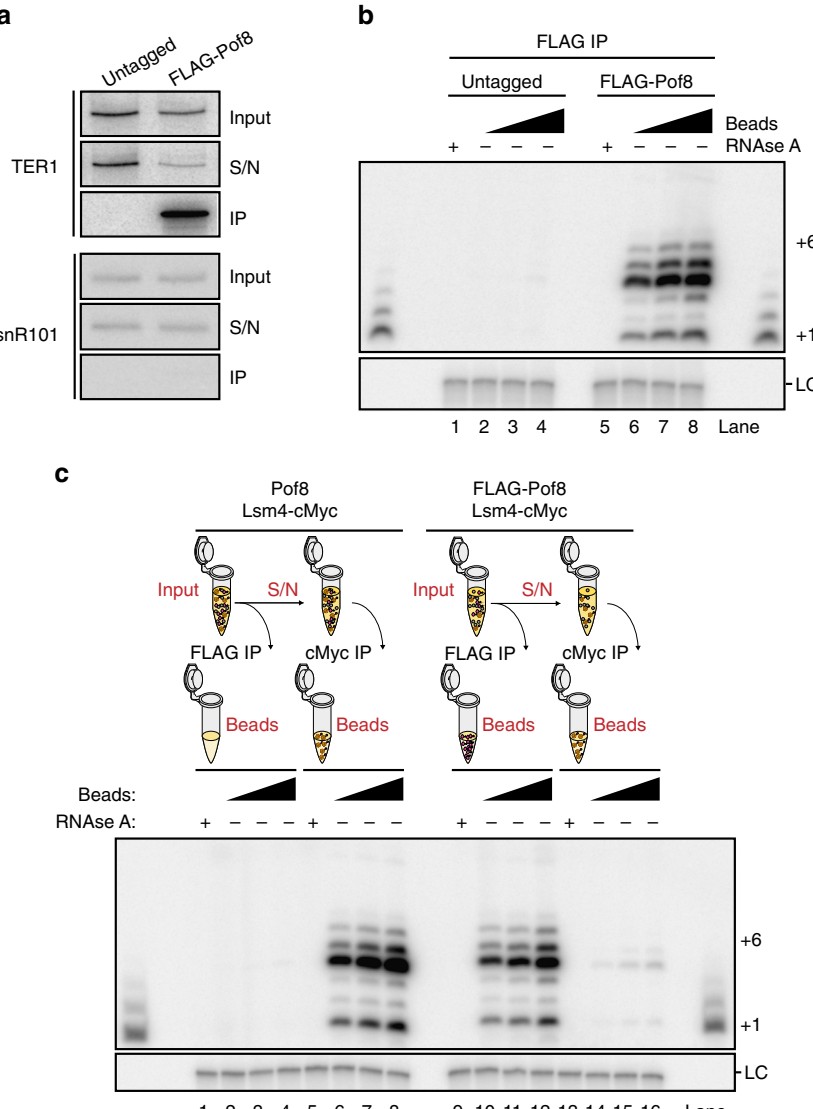

**Fig. 6** Pof8 is a constitutive member of telomerase holoenzyme. **a** Northern blot analysis of TER1 from α-FLAG immunoprecipitation (IP) from Pof8 and FLAG-Pof8 strains. Input and supernatant (S/N) represent 10% of the IP samples. snR101 was used as a negative control. The uncropped blots are presented in Supplementary Fig. 3d. **b** Telomerase activity assay from FLAG IP samples of FLAG-Pof8 and untagged Pof8 strains in an Lsm4-cMyc background. RNAse A was added to beads before performing telomerase assays in control samples. A ladder of extension products generated by terminal transferase was run on both edges of the gel; the bands corresponding to +1 and +6 nucleotide addition products are indicated on the right. A $^{32}$P-labeled 100-mer oligonucleotide was used as LC. **c** Telomerase assay of samples generated by sequential IPs with FLAG and cMyc antibodies. A schematic of the experiment is shown above the blot: first, extracts were exposed to α-FLAG antibody-bound beads, the supernatant (S/N) from the α-FLAG IP was then exposed to α-cMyc antibody-bound beads followed by a telomerase assay for all groups of IP samples. The uncropped blot is presented in Supplementary Fig. 3d

dodecyl sulfate (SDS) was then added to a final concentration of 2% (w/v) and incubated for 10 min at 65 °C. Then 5× TE (50 mM Tris-HCl pH 8.0, 5 mM EDTA) was added to a final volume of 10 ml and proteinase K (Sigma-Aldrich, P2308) to a final concentration of 50 µg ml$^{-1}$. After incubation for 1 h at 50 °C, the samples were precipitated with 3 ml of 5 M potassium acetate for 30 min on ice. The precipitates were removed with two rounds of centrifugation at 3200 × g for 10 min at 4 °C. The supernatant was collected and mixed with 1 volume of 100% isopropanol for 1 h on ice followed by centrifugation at 10,500 × g for 10 min at 4 °C. Genomic DNA was resuspended in 5× TE with 50 µg ml$^{-1}$ RNAse A. Resuspended DNA was then incubated for 1 h at 37 °C followed by two rounds of extraction with phenol: chloroform:isoamyl alcohol (25:24:1, equilibrated with 5× TE) and one round of chloroform:isoamyl alcohol (24:1, equilibrated with 5× TE). DNA was ethanol precipitated and resuspended in 1× TE. DNA concentrations were determined on a Qubit 3.0 instrument using the dsDNA BR Assay Kit (Life Technologies, Q32853) and 750 ng of each sample was digested with EcoRI for 12 h and then loaded onto a 1% agarose gel. The digested DNA was electrophoresed in 0.5× TBE (44.5 mM Tris-borate, 1 mM EDTA at pH 8.3) at 120 V for 6 h. Gels were stained with 1 µg ml$^{-1}$ ethidium bromide and visualized with Typhoon 8600 scanner to confirm

digestion of loaded DNA. Gels were then incubated in 0.25 M hydrochloric acid for 10 min followed by 0.5 M sodium hydroxide and 1.5 M sodium chloride buffer for 30 min and 0.5 M Tris-HCl (pH 7.5) and 1.5 M sodium chloride for 30 min at room temperature. DNA was transferred to Amersham Hybond-N+ membrane (GE Healthcare Life Sciences) via capillary blotting. Transferred DNA was cross-linked to the membrane in a Stratalinker using a 254-nm UV light at 120 mJ cm$^{-2}$. A probe specific for telomeric sequences was generated by PCR from pTELO using T3 (5′-ATTAACCCTCACTAAAGGGA-3′) and T7 (5′-TAATACGACTCACTA-TAGGG-3′) oligos. A probe specific for the rad16 gene was generated by PCR from wild-type genomic DNA using primers XWP9 (5′-ATGGTATTTTTTCGCCATT TACTCG-3′) and XWP10 (5′-TAGGCGGATCGTGAAGTTAA-3′). Both probes were labeled by random hexamer labeling with High Prime (Roche, 11585592001) and [α-$^{32}$P]-dCTP. Hybridizations were carried out with 10 million counts per minute of probe in Church–Gilbert buffer[45] at 65 °C. Blots were exposed to PhosphorImager screens and visualized with a Typhoon 8600 scanner.

To amplify chromosome end fusions, PCR reactions (25 µl) contained 1× ThermoPol buffer (NEB), 200 µM dNTPs, 0.5 µM of Bloli1256 (GGGTTGCAAA GTATGATTGTGGTAA), and Bloli1353 (TGTTGAATGTCAGAACCAACTGTT

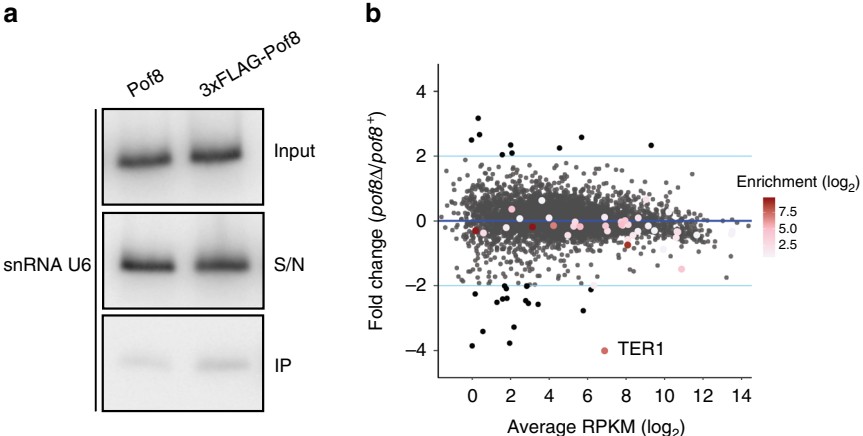

**Fig. 7** Pof8 is not a general loading factor for Lsm2-8. **a** Northern blot for U6 snRNA from input (10%), supernatant (S/N, 10%), and immunoprecipitates (IP, 100%) with anti-FLAG from extracts of cells expressing untagged or FLAG epitope-tagged Pof8. **b** Scatter plot relating changes in gene expression upon deletion of *pof8* to enrichment of RNAs in Lsm8 immunoprecipitate. Gene expression in $\log_2$ average RPKM is plotted on the x-axis and the $\log_2$ fold change between *pof8Δ* and wildtype on the y-axis. Differentially expressed genes with an absolute $\log_2$ fold change of ≥1 and an adjusted p-value < 0.05 are colored black. Genes that are enriched in the Lsm8 IP are colored by $\log_2$ (Lsm8 IP/Control IP) as indicated by the color gradient in the legend

GCAT) to amplify fusion junctions, 0.1 μM of Bloli3400 (GCAAAGAAGTTTCC TGGAATAGC) and Bloli3405 (GATGTAATAAAGGGTCGGCAC) to amplify part of the *trt1* gene as loading control, 1.25U Taq polymerase (NEB, M0273) and 1 ng of genomic DNA. Reactions were incubated at 95 °C for 30 s, followed by 32 cycles of 95 °C for 15 s, 55 °C for 30 s, and 68 °C for 3 min with a final extension at 68 °C for 10 min.

**Native protein extract and immunoprecipitation**. Cultures (2 l) were grown to a density of 0.5–1 × 10⁷ cells per ml and harvested by centrifugation for the preparation of cell-free extract[3,44]. Cells were washed three times with ice-cold TMG (300) buffer (10 mM Tris-HCl pH 8.0, 1 mM magnesium chloride, 10% (v/v) glycerol, 300 mM sodium acetate), and resuspended in two packed cell volumes of TMG(300) plus complete EDTA-free protease inhibitor cocktail (Roche), 0.5 mM PMSF, 1 mM EDTA, and 0.1 mM DTT and quick-frozen by dripping the cells suspension in small droplets into liquid nitrogen. Cells were lysed in a 6850 Freezer mill (SPEX SamplePrep) using eight cycles (2 min) at a rate of 10 per second with 2 min cooling time between cycles. Lysates were thawed on ice and one additional packed cell volume of TMG(300) plus supplements was added. Lysates were then cleared by centrifugation twice for 10 min at 6000 × g in a Beckman JA-17 rotor and then once for 45 min in a Beckman 70Ti rotor at 36,000 × g. All steps were carried out at 4 °C. Protein concentration was determined by Bradford assay and ranged between 6 and 11 mg ml⁻¹.

For RNA IPs, extracts (5.5 mg) were diluted to 5 mg ml⁻¹ with TMG(300) buffer plus supplements. An aliquot (100 μl) was frozen as input control. Heparin was added to 1 mg ml⁻¹ and Tween-20 to 0.1% (v/v). Magnetic dynabeads protein G (30 mg ml⁻¹; Invitrogen) was coated with anti-c-Myc 9E10 or anti-FLAG M2 (10 μg per 50 μl of bead suspension; Sigma-Aldrich, M4439 and F3165) by incubation for 30 min at room temperature in 200 μl of 1× PBS + 0.1% (v/v) Tween-20. Beads were washed three times with 1 ml of TMG(300). Immunoprecipitation was performed with 60 μl (Pof8) and 120 μl (Sm/Lsm) of bead suspension for 4 h at 4 °C with gentle rotation. Beads were collected using a magnet and an aliquot (100 μl) of supernatant was removed and frozen for further analysis. The beads were then washed five times with 1 ml TMG(300) plus supplements and 0.1% (v/v) Tween-20, once with TMG(200) (as TMG(300) except sodium acetate was at 200 mM) plus supplements and 0.1% (v/v) Tween-20 and once with TMG(50) plus supplements. Finally, beads were resuspended in 120 μl TMG(50) plus supplements and 0.4 U μl⁻¹ RNAsin (Promega) and frozen in liquid nitrogen.

**RNA preparation**. For total RNA extraction, cells (500 ml) were grown to a density of 5 × 10⁶ cells per ml and collected by centrifugation, washed twice with ddH₂O (500 ml), resuspended in 3 ml ddH₂O and quick-frozen by dripping the cells suspension in small droplets into liquid nitrogen. Cells were lysed in 6850 Freezer mill (SPEX SamplePrep) using seven cycles (2 min) at a rate of 10 per second with 2 min cooling time between cycles. The lysed cells were transferred into 50 ml tubes containing 10 ml phenol:chloroform:isoamyl alcohol (25:24:1, equilibrated with 50 mM sodium acetate, pH 5.2) and 10 ml 50 mM sodium acetate and 1% (w/v) SDS preheated to 65 °C. RNA was extracted four times with 10 ml phenol: chloroform:isoamyl alcohol (25:24:1, equilibrated with 50 mM sodium acetate, pH 5.2) and once with chloroform:isoamyl alcohol (24:1, equilibrated with 50 mM sodium acetate, pH 5.2). Total RNA was ethanol precipitated and resuspended in ddH₂O.

In the context of immunoprecipitation experiments, RNA was isolated from input, supernatant, and beads by incubation with proteinase K (2 μg μl⁻¹ in 0.5% (w/v) SDS, 10 mM EDTA pH 8.3, 20 mM Tris-HCl pH 7.5) at 50 °C for 15 min, followed by extraction with phenol:chloroform:isoamyl alcohol and chloroform: isoamyl alcohol. RNA was ethanol precipitated for 4 h at −20 °C and resuspended in ddH₂O. RNA used for RT-PCR was further DNase treated using the RNeasy Mini Kit (Qiagen) following the manufacturer's instructions.

**Northern blot analysis**. Where indicated, RNaseH cleavage was carried out on 15 μg of DNAse-treated total RNA isolated from *S. pombe*. RNA was combined with 600 pmol of BLoli1043 (5′-AGGCAGAAGACTCACGTACACTGAC-3′) and BLoli1275 (5′-CGGAAACGGAATTCAGCATGT-3′) targeting exon 1 and exon 2, respectively. The mixture was heated to 65 °C in a thermocycler for 5 min and then allowed to slowly cool down at room temperature for 10 min. About 1× RNaseH buffer (NEB) and 5U RNaseH enzyme (NEB, M0297) were added to the mixture and incubated for 30 min at 37 °C. RNaseH-treated samples were ethanol precipitated for 4 h at −20 °C and centrifuged at 2000 × g for 20 min at 4 °C. RNA was then resuspended in 1× formamide loading buffer and separated on a 4% (v/v) polyacrylamide (29:1) gel containing 8 M urea and transferred to Biodyne nylon membrane (Pall Corporation) at 400 mA for 1 h in 0.5× TBE buffer. RNA was crosslinked to the membrane using 254-nm UV light at 120 mJ cm⁻² in Stratalinker (Stratagene). Hybridization with radiolabeled probes (10 million counts per minute) were performed in Church–Gilbert buffer[45] at 60 °C for TER1 probe (nucleotides 536–998, labeled with High Prime (Roche) and [α-³²P]-dCTP), and at 42 °C for small nucleolar RNA snRN101 and U6 snRNA (oligonucleotide BLoli1136 (5′-CGCTATTGTATGGGGCCTTTAGATTCTTA-3′) and BLoli4628 (5′-TCTGTATCGTTTCAATTTGACCAAAGTGAT-3′), respectively, labeled with T4 polynucleotide kinase (NEB, M0201) in the presence of [γ-³²P]-ATP). Blots were exposed to PhosphorImager screens and analyzed with a Typhoon 8600 scanner. The uncropped blots of this work are shown in Supplementary Fig. 3.

**Western blot analysis**. Western blot analysis was performed with native protein extracts, prepared as described above, diluted to 6 μg μl⁻¹ and mixed with equal volume of 2× protein sample buffer (2× NuPAGE LDS buffer (Life Technologies), 100 mM DTT, 4% (w/v) SDS). Samples were then incubated for 10 min at 75 °C and 10 μl (30 μg) of samples was loaded onto a 4–12% NuPAGE Bis-Tris gel (Life Technologies, NP0321BOX). Electrophoresis was done in 1× MOPS buffer (Life technologies, NP0001) at 160 V for 60 min. Proteins were transferred to Protran nitrocellulose membranes (Whatman) in western transfer buffer (3.03 g l⁻¹ Tris base, 14.4 g l⁻¹ glycine, 20% (v/v) methanol) at 100 V for 1 h. Blots were blocked and washed with iBind Flex Western Device (Life Technologies, SLF20002). Lsm and Sm blots were probed with mouse monoclonal anti-c-Myc 9E10 (Sigma-Aldrich, M4439) at 1:5000 dilution and horse-radish peroxidase-conjugated goat anti-mouse IgG (H+L) at 1:5000 (Thermo Scientific, 31430). Trt1 blot was probed with rabbit polyclonal anti-cMyc A14 (Santa Cruz Biotechnologies, sc-789) at 1:400 dilution and horse-radish peroxidase-conjugated goat anti-rabbit IgG (H+L) at 1:4000 (Thermo Scientific, 31460). Blots were reprobed with mouse anti-α-tubulin (Sigma-Aldrich, T5168).

**RT-PCR**. DNAse-treated RNA samples were used for RT-PCR reaction as describe[8]. Primers for RT reaction were BLoli1275 (5′-CGGAAACGGAATT-CAGCATGT-3′) for precursor and spliced form; PBoli918 (5′-

ACAACGGACGAGCTACACTC-3′) for first exon; and BLoli2051 (5′-GACCT TAGCCAGTCCACAGTTA-3′) for U1 as loading control. RNA samples (2.5 μg) were combined with oligos (10 pmol) and dNTP mix (10 nmol) in 13 μl, and samples were heated to 65 °C for 5 min. After cooling, the volume was increased to 20 μl by the addition of 40 U of RNasin (Promega), 5 mM DTT, 1× first-strand buffer, and 200 U of Superscript III reverse transcriptase (Invitrogen). Samples were incubated at 55 °C for 60 min. RNaseH (5 U, NEB, M0297S) was added followed by incubation at 37 °C for 20 min. Aliquots (2 μl) of the RT reactions were used for PCR amplification with Taq polymerase (NEB, M0273) under the following conditions: 5 min at 94 °C followed by 28 cycles of 30 s at 94 °C, 30 s at 57 °C, and 60 s at 72 °C, followed by 10 min at 72 °C. Primers used were BLoli1275 and Bloli1020 (5′-CAAACAATAATGAACGTCCTG-3′) for precursor and spliced form, PBoli918 and BLoli1006 (5′-CATTTAAGTGCTTGTCAGATCACAACG-3′) for first exon, and BLoli2051 and BLoli2101 (5′-ACCTGGCATGAGTTTCTGC-3′) for U1.

**Telomerase activity assay.** Telomerase was immunoprecipitated on magnetic dynabeads protein G (Invitrogen, 10003D) coated with anti-c-Myc 9E10 (Sigma-Aldrich, M4439) for Trt1 and LSm4, LSm5 or anti-FLAG M2 (Sigma-Aldrich, F3165) for Pof8 as described above. Three amounts of bead suspension (5, 10, and 20 μl) were used for the telomerase activity assay. Negative control samples were incubated in 20 μl of TMG(50) plus 20 ng of RNAse A (Invitrogen) for 10 min at 30 °C. Beads were incubated in 10 μl of 50 mM Tris-acetate at pH 8.0, 100 mM potassium acetate, 1 mM magnesium acetate, 5% (v/v) glycerol, 1 mM spermidine, 1 mM DTT, 0.2 mM dATP, dCTP, dTTP, 2 μM [α-$^{32}$P]-dGTP (500 Ci mmol$^{-1}$), and 5 μM of oligo PBoli871 (5′-GTTACGGTTACAGGTTACG-3′). Reactions were incubated for 90 min at 30 °C and stopped by the addition of proteinase K (2 μg μl$^{-1}$ in 0.5% (w/v) SDS, 10 mM EDTA pH 8.3, 20 mM Tris-HCl pH 7.5) plus 1000 cpm 100-mer labeled with [γ-$^{32}$P]-ATP as the loading control at 42 °C for 15 min. Primer extended products were extracted with phenol:chloroform:isoamyl alcohol (25:24:1, equilibrated with 5× TE) and ethanol precipitated for 4 h at −20 °C. Extracted DNA was electrophoresed in 10% (v/v) polyacrylamide (19:1) sequencing gel containing 8 M urea for 1.5 h at 80 W. Gels were dried and exposed to PhosphorImager screens, and analyzed with a Typhoon 8600 scanner.

**UV crosslinking and denaturing immunoprecipitation.** A Ter1 probe corresponding to the two short arms was generated by fusing an Sp6 promoter sequence with nucleotides +1 to +97 and +955 to +1212 of TER1 and a hepatitis δ virus (HDV) ribozyme sequence to allow for production of a precisely defined 3′ end. Primers used to generate the DNA template are listed in Supplementary Table 2. The TER1 +1 to +97 fragment with Sp6 promoter sequence was amplified from pJW10[3] with BLoli7098 (containing Sp6 promoter sequence) and BLoli7099 (containing 20 nt overlapping sequence with +955 to +1212 fragment). The TER1 +955 to +1212 fragment with HDV sequence (5′-GGGCGGCCATGGTCCCAGC CTCCTCGCTGGCGCCGCCTGGGCAACATGCTTCGGCATGGCGAATGGGA CCAA-3′) was PCR amplified from pTER1-i33 with primers Bloli7100 and Bloli6540. The entire probe sequence was then amplified by fusion PCR from the two fragments as template. PCR reactions (50 μl) contained 1× Phusion HF buffer (Life Technologies), 200 μM dNTPs, 0.5 μM of each primer, and 0.02 U μl$^{-1}$ of Phusion Hot Start II DNA Polymerase (Life Technologies, F549). Reaction conditions were: 98 °C for 30 s, followed by 32 cycles of 98 °C for 10 s, 65 °C for 30 s, and 72 °C for 30 s with a final extension at 72 °C for 10 min. The PCR product was cloned using Zero Blunt PCR Cloning Kit (Life Technologies, K270020) following manufacturer's instructions to give rise to plasmid pDP2 which was sequence verified. The template for in vitro transcription was generated by PCR amplification from plasmid pDP2 with primers BLoli7098 and BLoli6540 using the PCR conditions listed above.

The in vitro transcription reactions contained 1× transcription buffer (Promega), 0.5 mM each of ATP, CTP, GTP, 0.1 mM UTP, 0.66 μM [α-$^{32}$P]-UTP (3000 Ci mmol$^{-1}$), 1 μg DNA template, 40 U RNasin (Promega, N2111), and 1.9 U SP6 RNA polymerase (Promega, P1085) in a 10 μl volume. Reactions were incubated at 37 °C for 3 h (time optimized to maximize the amount HDV ribozyme cleaved product), and 2 U of DNase I (NEB, M0303) was added, followed by incubation for 30 min at 37 °C. Reactions were stopped by the addition of formamide loading dye. The full-length HDV-cleaved TER1 probe product was gel purified on 5% (v/v) polyacrylamide gels containing 8 M urea for 2 h at 15 W. The transcribed and processed RNA is predicted to fold into the same structure as in the context of the full-length RNA using mFOLD default parameters (http://unafold.rna.albany.edu/?q=mfold).

To capture proteins that directly interact with TER1, the RNA probe (2.0 nM) was incubated on ice for 30 min with native protein extracts from strains containing cMyc-tagged Lsm4, FLAG-tagged Pof8, and no tag, respectively. Reactions (100 μl) contained 1.5% (w/v) PEG8000 (NEB), 60 mM potassium phosphate pH 7.0, 1 mM spermidine, 2 mM MgCl₂, 0.4 U of RNAsin (Promega, N2111), and 40 μl of 8 μg μl$^{-1}$ native protein extract. After incubation, 50 μl of the reaction was aliquoted into two 25 μl droplets onto parafilm stretched over an aluminum block that was precooled at 4 °C and irradiated in a Stratalinker (Stratagene) using 254 nm UV light at 0.8 J cm$^{-2}$. After crosslinking, 1% (w/v) of SDS, 1% (v/v) of Triton X-100, and 100 mM DTT were added to the crosslink and no crosslink reactions, and then heated in boiling water for 2 min. Denatured

samples were diluted 10-fold with TMG(300) buffer and immunoprecipitated with magnetic dynabeads protein G (Invitrogen) coated with anti-c-Myc 9E10 or anti-FLAG M2 (Sigma-Aldrich, M4439 and F3165) antibody as described in native protein extract and immunoprecipitation section. After the immunoprecipitation, beads were washed four times with 1 ml TMG(300) and once with 1 ml TMG(50), and treated with proteinase K (2 μg μl$^{-1}$ in 0.5% (w/v) SDS, 10 mM EDTA pH 8.3, 20 mM Tris-HCl pH 7.5) at 42 °C for 15 min. The supernatant was extracted with phenol:chloroform:isoamyl alcohol (25:24:1, equilibrated with 50 mM sodium acetate, pH 5.2) and ethanol precipitated for 4 h at −20 °C. RNA was resolved on a 5% (v/v) polyacrylamide containing 8 M urea for 2 h at 15 W. Gels were dried onto Whatman 3MM Chr blotting paper (GE Healthcare Life Sciences) and exposed to a PhosphorImager screen for analysis with a Typhoon 8600 scanner.

**RNA-Seq and RIP-Seq.** Ribo-depleted stranded RNA-Seq libraries were constructed for three Pof8 samples and isogenic controls using the TruSeq SBS Kit v4-HS kit (Illumina). The libraries were pooled and sequenced on an Illumina HiSeq 2500 for 100-bp single-end reads on two lanes. Read counts per library ranged from 56 million to 84 million. Reads were aligned to the S. pombe ASM294v2 genome from ensembl using the STAR aligner[46] (v.2.5.2b) with the following parameters: -outFilterType BySJout --alignSJDBoverhangMin 5 --alignSJoverhangMin 10 --alignIntronMin 20 --alignIntronMax 2500 --twopassMode Basic. Between 49 and 72 million reads per library passed filtering. A counts table for unambiguous uniquely mapped reads was generated using a custom R script. Coordinates for snu4 (II: 467,488–467,615)[47] and snu5 (II: 3,236,356–3,237,312) (GenBank: X16573.1) were manually curated based on alignment of the published sequences to the genome. Differential gene expression analysis was performed with EdgeR[48] (v.3.14.0) using the likelihood ratio test with the Benjamini–Hochberg false discovery rate correction. Genes with fewer than one count per million in three or more libraries were filtered prior to differential expression analysis. Genes with absolute log₂ fold change >1 with an adjusted p-value < 0.05 were considered differentially expressed.

RNA libraries from Lsm8 and control immunoprecipitations were prepared following instructions for Solexa messenger RNA sequencing and each sample was sequenced on a single lane of the Illumina GAIIx sequencer producing 40 bp single-end unstranded reads. The fastq files were aligned to the S. pombe reference genome ASM294v2 using the STAR aligner (v 2.5.2b) with the following parameters: --outFilterType BySJout --outFilterMultimapNmax 20 --alignSJoverhangMin 8 --alignSJDBoverhangMin 1 --alignIntronMin 20 --alignIntronMax 1000 --outSAMtype BAM SortedByCoordinate --twopassMode Basic. The corresponding reference annotation gtf file was manually curated for snu4 using coordinates II:467,488–467,615. A counts table for unambiguous uniquely mapped reads was generated using a custom R script. There were 942,516 read counts for the control IP and 6,604,836 reads for the Lsm8 IP sample. The read counts for each gene were normalized to the median read counts per library. Genes were called enriched if the normalized counts in the IP sample exceeded the corresponding counts in the control by 10 or more and the normalized read count ratio was >2.

**Data availability.** Original data underlying this manuscript can be accessed from the Stowers Original Data Repository at http://www.stowers.org/research/publications/LIBPB-1190. Gene expression data and immunoprecipitation data have been deposited in NCBI's Gene Expression Omnibus (GEO) database under accession number GSE104672.

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

## Acknowledgements

We thank Li Chen for plasmid pTER1-i33, Katie Hildebrand for technical assistance, Kristi Jensen and Lisa Lassise for help with preparing the manuscript, and Chi-Kang Tseng and other members of the Baumann laboratory for discussions. We also thank Laura Collopy, Kazu Tomita, and Toru Nakamura for sharing data prior to publication. This work was funded in part by the Stowers Institute for Medical Research and was performed in part to fulfill, in part, requirements for D.J.P.-M.'s thesis research in The Graduate School of the Stowers Institute for Medical Research. P.B. is an Investigator with the Howard Hughes Medical Institute.

## Author contributions

D.J.P.-M. and P.B. designed the study, D.J.P.-M. carried out most of the experiments except for data presented in Figs. 2d and 3c (L.P.) and part of Fig. 6b (W.T.). R.F.S. processed the *pof8* expression data and M.R.S. processed the RNA IP data set. D.J.P.-M. and P.B. analyzed the data and prepared the manuscript.

## Additional information

**Competing interests:** The authors declare no competing financial interests.

