## [Peer Review File(PDF 148 kb) · Nature Communications]

Reviewers' comments:

Reviewer #1 (Remarks to the Author):

The paper by Paez-Mscoso et al identifies Pof8 as component required for telomerase holoenzyme assembly. This is an exciting paper, with a clear logic and is experimentally very well done. Briefly, Paez-Mscoso et al purify LSM2 and LSM8 associated proteins and identify Pof8 as an interacting factor. Based on the RRM and La motives in Pof8, a similarity to p65, the authors test the role of Pof8 in the telomerase biogenesis pathway. Pof8 deletes have shorter telomeres, lower total TER1 levels and severely reduced telomerase activity. Through several independent experiments the authors show that the reduced telomerase expression can not be explained by the reduced TER1 levels alone. Instead elegant direct and sequential pull down experiments of TER1 with endogenously tagged LSM and Pof8 demonstrate that Pof8 associates with the LSM bound, active form of telomerase. Moreover, the authors show that loss of Pof8 leads to an accumulation of Smb associated precursor TER1 complexes and a reduction in the active LSM bound form. This work uncovers an additional level of complexity in telomerase maturation pathway. This work is a perfect fit for the broad readership of Nature Communications and I have only minor suggestions.

Minor comments:

- In figure 4a there are two TER1 bands in the input and S/N but only one of the band is IPed. It would be helpful to the reader to explain why there are two bands and how the upper band was considered in the quantification.
- The discussion is well written and highlights the finding of the paper in the relevant context of assembly pathways in different organisms. The reader would also benefit from a more detailed discussion of the last sentence in the paper. It would be helpful if the authors would explain in a couple of sentences how they think about the order of events taking place during telomerase assembly.
- Have the authors considered renaming Pof8 to a Lar protein?

Reviewer #2 (Remarks to the Author):

In the two dual-submission manuscripts by Paez-Moscoso et al and Collopy et al, the groups identify a new key member of the telomerase enzyme in fission yeast, termed respectively Pof8 and Lar7. Collopy et al first identify Pof8, a putative f-box protein, from a previous study (Liu et al) as a cause of a very short telomere phenotype. Because of domain similarity to the LARP7 protein family, Collopy et al rename the protein to Lar7. Paez_Moscoso et al first identify Pof8 while performing immunoprecipitation experiments from Lsm proteins (results in preparation for additional manuscript). They too note its similarities to LARP7 protein family.

In fission yeast, the RNA component TER1 is generated initially by an incomplete spliceosomal cleavage reaction. It is first bound by Sm proteins and later by Lsm proteins and Trt1, the catalytic subunit.

To show that a protein is a core component of the telomerase enzyme, the authors should demonstrate that loss of the protein impairs telomere maintenance. They should also show that this protein can precipitate the other core telomerase components including the RNA component (TER1) and be able to precipitate a majority of telomerase activity. They should also investigate the effect of loss of the protein on stability of the telomerase enzyme and its components and the

effect on telomerase activity. Finally, they should look at which components of the telomerase complex this new protein is binding to.

Paez-Moscoso: In this manuscript from Peter Baumann's lab, the authors very nicely show that Pof8 is required for TER1 stability. However, by overexpressing TER1 they are able to additionally demonstrate that Pof8 is required for telomerase assembly and enzyme activity independent of TER1 levels. They show that pof8 binds TER1 and can precipitate a majority of telomerase activity. They show that in the absence of pof8, Lsm proteins don't assemble with TER1. By comparing RNA co-precipitating with Lsm and differentially expressed genes upon pof8 deletion, they show that TER1 is unique in its requirement for pof8 for stability and loading into Lsm complexes.

Minor Concerns:

- In figure 2b, RT-PCR is performed on different TER1 species and the products run on a gel. qRT-PCR would be more quantitative.
- Most of the northern blots are quantified, except for figure 3a.
- In figure 3c, should we see telomere elongation upon induction of ter1 expression?
- Figure 6a missing labels- its not clear that this is a blot for U6

Major Concerns

- In figure 5c, the authors show that they can immunoprecipitate telomerase activity using Pof8, and in fact deplete the majority of telomerase activity. However, they don't demonstrate pulldown of Trt1 or Lsm proteins using Pof8. Because of this, they can't show which component Pof8 is binding to (although it's likely TER1, due to the multiple RNA binding motifs.)
- There are no western blots for untagged telomerase enzyme components (Lsm, Sm, Trt1) to show that their levels are unaffected by Pof8 deletion
- In figure 4a, the authors show more TER1 bound to Sm proteins in the absence of Pof8. This stands in contradiction to data in the Collopy paper (see figure 5a).

October 9, 2017

Manuscript ID: NCOMMS-17-16998A

Response to reviewers' comments:

We thank reviewer 1 for referring to our work as “an exciting paper, with a clear logic and [...] experimentally very well done”. Such comments are very motivating to a graduate student receiving the reviews for his first paper and I appreciate the reviewer taking the time to write them. We are also delighted that the reviewer thinks that “*This work is a perfect fit for the broad readership of Nature Communications*”

- *In figure 4a there are two TER1 bands in the input and S/N but only one of the band is IPed. It would be helpful to the reader to explain why there are two bands and how the upper band was considered in the quantification.*

The two bands in Figure 4a most likely represent precursor and mature form. Consistent with **new Figure 2a**, where the bands are more clearly separated following RNaseH cleavage, the abundance of precursor is largely unaffected by deletion of *pof8*, whereas the mature form (lower band) is reduced. This is also seen in figure 4a in input and supernatant (lower band is reduced in *pof8*Δ samples). For quantification purposes, we do not distinguish between the two bands in figure 4a as they run close together in the input and are difficult to quantify separately. The enrichment shown below the lane is relative to total TER1 from the input. This is now stated in the Figure legend. Following RNase H cleavage, a better separation is achieved and the bands have been quantified separately in figure 2a.

- *The discussion is well written and highlights the finding of the paper in the relevant context of assembly pathways in different organisms. The reader would also benefit from a more detailed discussion of the last sentence in the paper. It would be helpful if the authors would explain in a couple of sentences how they think about the order of events taking place during telomerase assembly.*

In response to the reviewer's suggestion we have expanded the discussion to further elaborate on the order of events during hierarchical assembly.

- Have the authors considered renaming Pof8 to a Lar protein?

We have indeed considered referring to Pof8 by a new name to indicate the structural and functional similarities we have identified. We decided against it for the following reasons: Firstly, nomenclature rules for *S. pombe* require new names to conform to the three-letters-plus-integer pattern. Names should stand for a description of a phenotype, gene product or gene function. As the gene in question was already named Pof8 several years ago, we would have to register an additional name with the naming committee, a practice that is discouraged by the naming committee. To our knowledge,

Dr. Tomita is taking this route and has reserved the name lar7 for La related protein 7 and refers to Pof8 by this new name. This name will be added to the database and researchers are encouraged to resolve this conflict in the literature in the future.

It seems far from certain to us that Pof8 is a functional ortholog of mammalian LARP7 as suggested/implied by the name lar7. There is a danger here of replacing one name we deem to be incorrect, as we were unable to identify the previously reported F-box in (Pombe F-box protein 8), with another name that implies a functional similarity that may or may not exist. LARP7 has been implicated in regulating TERT splicing, but not in hTR biogenesis. Clearly, we and the Tomita group have revealed functional similarities between Pof8 and p65 and p43 in ciliates, but these proteins were named based on molecular weight estimates, as common among biochemists. Such names do not conform to the *S. pombe* naming standards. Structural domain similarity and the implication of LARP3 in hTR biogenesis may justify the name lar3. However, human LARP3 is the bona fide La protein and may thus be the functional ortholog of sla1 (S. pombe La protein). For those reasons, we prefer to only advocate change of a published name after further studies have identified clear structural or functional orthologies between the pombe and mammalian family members.

Reviewer #2

We agree with reviewer 2 regarding all standards he/she lays out to show that a protein is a core component of the telomerase enzyme. We believe that these are all addressed in the revised manuscript. We have inserted references to the Figure panels that address the point in question:

They should also show that this protein can precipitate the other core telomerase components including the RNA component (TER1) (Figure 5a) and be able to precipitate a majority of telomerase activity (Figure 5b and c). They should also investigate the effect of loss of the protein on stability of the telomerase enzyme and its components (Figure 2a and new Suppl. fig. 1) and the effect on telomerase activity (Figure 2e). Finally, they should look at which components of the telomerase complex this new protein is binding to (new Figure 5c).

- In figure 2b, RT-PCR is performed on different TER1 species and the products run on a gel. qRT-PCR would be more quantitative.

We have now replaced Figure 2a with a Northern blot following RNaseH cleavage which allows for good separation (and thus independent quantification) of the precursor and mature forms of TER1. Direct quantification on Northern blots confirms our previous results that the mature form is 3-4 fold reduced in the absence of Pof8, whereas the precursor remains largely unchanged. The semi-quantitative PCR in Figure 2b also

confirms this result and also shows that there is no apparent difference in the level of spliced TER1. As neither of these assays indicate a change in precursor or spliced form, we decided not to embark on optimizing qPCR for the different forms of TER1 to further support a negative result.

- Most of the northern blots are quantified, except for figure 3a.

The Northern blot in old figure 3a contained a transfer blemish affecting the TER1 band in lane 6. This Northern has now been replaced with a new one and quantifications are shown below the lanes.

- In figure 3c, should we see telomere elongation upon induction of ter1 expression?

We do not observe telomere elongation when overexpressing TER1 without also overexpressing Trt1, at least not on the scale of tens of population doublings. The Lingner lab reported overexpressing TR in human cells and observed telomere elongation of ~0.3% per population doubling. Such a modest increase would not be visible on our Southern blots.

- Figure 6a missing labels- its not clear that this is a blot for U6

The blot has now been labelled to make it clear that probing is for snRNA U6.

- In figure 5c, the authors show that they can immunoprecipitate telomerase activity using Pof8, and in fact deplete the majority of telomerase activity. However, they don't demonstrate pulldown of Trt1 or Lsm proteins using Pof8. Because of this, they can't show which component Pof8 is binding to (although it's likely TER1, due to the multiple RNA binding motifs.)

The fact that pulldown of Pof8 precipitates telomerase activity strongly argues that the catalytic subunit is precipitated alongside TER1 and Pof8. To address the concern that we had not shown whether Pof8 interacts directly with TER1 or binds a protein subunit, we have now directly UV-crosslinked TER1 to its binding partners, followed by immunoprecipitation under denaturing conditions. This experiment confirmed that Pof8 indeed interacts directly with TER1. These results are included as new Figure 5c and are described in the text.

- There are no western blots for untagged telomerase enzyme components (Lsm, Sm, Trt1) to show that their levels are unaffected by Pof8 deletion

To our knowledge there are no antibodies available against any of the components listed above. Examining the level of untagged proteins would thus be difficult. We have actually tried to raise anti Trt1 antibodies on several occasions without success. It is not clear to us why the reviewer would want to see Western blots of untagged proteins. In the absence of evidence that the tags affect protein stability, we have performed Western blots for tagged Lsm4, Lsm5, Smb1, Sme1 and Trt1 from strains deleted for pof8 and isogenic controls. No change was observed for Lsm and Sm

proteins and Trt1 levels were only slightly reduced. This data has been added as Suppl. Figure 1.

- *In figure 4a, the authors show more TER1 bound to Sm proteins in the absence of Pof8. This stands in contradiction to data in the Collopy paper (see figure 5a).* Unfortunately, the nature of this contradiction is not apparent to us as we have not seen a copy of their manuscript. Did they show that the same amount of TER1 is precipitated with Sm proteins in the presence and absence of Pof8? Could this be a question of the sensitivity of the assay? We stand by the result that more TER1 is precipitated by Sm proteins in the absence of pof8 after normalization for the differences in input due to the lower steady-state level of TER1 in pof8 Δ cells. This has been reproduced in independent experiments using extracts from cells with tags on different Sm proteins. The result is also consistent with Tang et al. 2012 showing that a TER1 mutant that fails to bind Lsm2-8 shows increased binding for Sm. It is important to note though that in both cases the apparent functional consequences relate to the dramatic decrease in Lsm binding, not the slight increased Sm binding.

REVIEWERS' COMMENTS:

Reviewer #1 (Remarks to the Author):

All my concerns have been addressed, please publish as is.

Reviewer #2 (Remarks to the Author):

The authors have addressed all my critique points.